# Time to Diagnosis and Treatment of Diabetes Mellitus among Korean Adults with Hyperglycemia: Using a Community-Based Cohort Study

**DOI:** 10.3390/ijerph191912090

**Published:** 2022-09-24

**Authors:** Ihn-Sook Jeong, Chan-Mi Kang

**Affiliations:** 1College of Nursing, Pusan National University, Yangsan 50612, Korea; 2Department of Nursing, Dong-Eui Institute of Technology, Busan 47230, Korea

**Keywords:** diabetes mellitus, diabetes complications, early diagnosis, time-to-treatment

## Abstract

Objectives: To identify the time from hyperglycemia to diabetes mellitus (DM) diagnosis and treatment, the risk factors for diabetes development, and the prevalence of comorbidities/complications in patients > 40 years of age. Methods: This secondary data analysis study used data from the Korean Genome and Epidemiology Study. The participants comprised 186 patients who did not have diabetes at baseline, but developed hyperglycemia at the first follow-up. The average and median periods until DM diagnosis and treatment were calculated using Kaplan–Meier survival analysis. Results: Of the 186 participants, 57.0% were men and 35.5% were 40–49 years old. The average time to DM diagnosis and treatment was 10.87 years and 11.34 years, respectively. The risk factors for the duration of DM were current smoking, body mass index (BMI), fasting blood sugar (FBS), and postprandial 2-hour glucose (PP2). The risk factors for the duration of diabetes treatment were current smoking, hypertension, BMI, FBS, and PP2. The development of one or more comorbidities or diabetes complications was identified at the time of DM diagnosis (36.5%) and DM treatment (41.4%). Conclusions: As diabetes complications occur at the time of DM, and early treatment can impact the development of diabetes complications or mortality, it is necessary to establish a referral program so that participants presenting with high blood sugar levels in the screening program can be diagnosed and treated in a timely manner.

## 1. Introduction

Diabetes mellitus (DM) is a hyperglycemic state caused by insulin resistance and deficiency due to genetic or environmental factors [1]. The number of people with DM worldwide is expected to increase to 700 million (10.9% of the adult population) by 2045 [2]. In 2019, the prevalence of DM in Korean adults aged ≥ 30 years was 14.5% [3]. According to national health insurance data, the cost of DM treatment in 2019 was KRW 2.7 trillion, a 17-fold increase compared to KRW 160 billion in 2002 [4]. The burden of DM ranks first in Korea’s disability-adjusted life years [5]. The economic burden of DM is expected to increase considering the rapid increase in the number of older adults with DM owing to the continuous aging of the population.

Hyperglycemia is the most common characteristic of both type 1 and type 2 DM and can cause serious complications owing to gradual and chronic effects on the human body [6,7]. Prolonged hyperglycemia leads to oxidative damage at the DNA, protein, and lipid levels, which causes cell necrosis or apoptosis [8]. Additionally, hyperglycemia damages the insulin signal transduction pathway, which increases glucose uptake by fat or muscle cells and decreases glucose synthesis in the liver. As these mechanisms cause cellular pathological damage and microvascular and macrovascular complications, early detection and effective management of hyperglycemia, including drug treatment, are necessary for prevention [6,7]. 

Diagnosis of DM is important for successful management; often, people with hyperglycemia are diagnosed with DM by a specialist who then initiates treatment [9]. Accordingly, the 2021 integrated health promotion project “prevention and control of cardiovascular diseases” [10] set the primary goal of improving the treatment rate of DM and DM awareness, and the secondary goal of increasing the DM control rate, increasing the symptom recognition rate for acute myocardial infarction or stroke, and decreasing the fatality rate. 

Even following a diagnosis of DM, the risk of microvascular and macrovascular complications increases as the period from hyperglycemia to DM diagnosis increases [11]. Thus, it is important for DM to be diagnosed and treated in patients with hyperglycemia as early as possible. Therefore, it is necessary to study the duration until the diagnosis or treatment of DM in patients with hyperglycemia, and the prevalence of microvascular and macrovascular complications at the time of diagnosis and treatment. The current study was conducted to determine the DM diagnosis and treatment rate, the time from hyperglycemia to DM diagnosis and treatment, and the prevalence patterns by selecting participants with persistent hyperglycemia among adults >40 years of age using a representative large-scale community-based cohort in Korea.

## 2. Methods

### 2.1. Participants

This study was conducted using data from the Korean Genome and Epidemiology Study (KoGES)-Ansan and Ansung cohorts. The collected KoGES data included core questionnaires on general characteristics, medical history, smoking and drinking status, and women’s health, and anthropometric and clinical measurements (blood and urine tests) [12,13]. This study used the combined data of prospective follow-ups from baseline to the 8th follow-up (2017–2018). The participants were adults aged 40–69 years who met all of the following selection criteria: (1) no hyperglycemia at baseline but hyperglycemia in the 1st follow-up; (2) neither diagnosis nor treatment of DM at the baseline and the 1st follow-up; (3) no underlying diseases at baseline; (4) no discrepancy between the time of diagnosis and treatment of DM; and (5) completion of at least one follow-up so that the diagnosis and treatment of DM could be confirmed in the follow-up. In this study, hyperglycemia was defined as abnormally high blood sugar in one or more of three glycemic indices, with glycated hemoglobin (HbA1C) ≥ 6.5%, fasting blood sugar (FBS) ≥ 126 mg/dL, or postprandial 2-hour glucose (PP2) level ≥ 200 mg/dL [14,15,16,17]. Underlying diseases included cerebrovascular disease, coronary artery disease, myocardial infarction, or kidney disease [6]. Among the 10,030 participants, we excluded patients who had hyperglycemia from the baseline and those who did not have hyperglycemia in the first follow-up (*n* = 9713). Additionally, those who were diagnosed with or treated for diabetes at baseline and the first follow-up (*n* = 49), participants with underlying diseases at baseline (*n* = 23), discrepancies in which the DM treatment time was earlier than diagnosis (*n* = 36), and participants without follow-up (*n* = 23) were excluded. Finally, a total of 186 participants were included in this study (Figure 1).

### 2.2. Variables and Definitions

#### 2.2.1. Participant Characteristics

The variables in this study were characteristics reported in previous studies to affect the diagnosis [18,19,20] and treatment of DM [9,18]. The variables were used as they were, or by reclassifying the KoGES data. General characteristics included sex, age, current drinking, current smoking, physical activity, comorbidities (hypertension and dyslipidemia), family history of DM, and clinical tests (body mass index (BMI), FBS, and PP2). The collected data were then reclassified based on the study objectives. The BMI was calculated using the following formula: weight (kg)/height^2^(m^2^). Based on the classification of the World Health Organization Asia-Pacific guidelines [21], BMI was categorized as “<25 kg/m^2^ (normal or overweight)” and “≥25 kg/m^2^ (obese)”. HbA1C was classified as “<6.5%” and “≥6.5%,” FBS was categorized as “<126 mg/dL” and “≥126 mg/dL,” and PP2 was classified as “<200 mg/dL” and “≥200 mg”/dL” [14,15,16,17].

#### 2.2.2. Time to Diagnosis and Treatment of DM

DM diagnosis was classified as “yes” or “no” according to the doctor’s diagnosis [18,19,22,23]. DM treatment was classified as “yes” or “no” depending on whether insulin treatment or oral antidiabetic medications were currently used for blood glucose control [18]. The period (years) from hyperglycemia to DM diagnosis or treatment was defined as the date of DM diagnosis or treatment from the date of hyperglycemia at the 1st follow-up. The DM diagnosis date was the follow-up date when DM was diagnosed, while the DM treatment date was the follow-up date when DM was treated. If the participants did not have a DM diagnosis or treatment, the follow-up period was considered the last follow-up date.

#### 2.2.3. Prevalence of Comorbidities and Complications of DM at Diagnosis and Treatment of DM

The prevalence of comorbidities was defined as an affirmative response to the survey question on the history of diagnosed hypertension or dyslipidemia and categorized into “yes” or “no”. Complications of DM were defined as an affirmative response to the survey question on the history of diagnosed cerebrovascular disease, coronary artery disease, myocardial infarction, or kidney disease at the time of diagnosis or treatment of DM [6] and categorized into “yes” or “no”.

### 2.3. Data Analysis

Data were analyzed using SPSS (version 26.0; IBM Corp., Armonk, NY, USA), and statistical significance (α) was set at *p* < 0.05. The characteristics of the participants in the first follow-up and the diagnosis and treatment rates of diabetes during the entire period were calculated as the frequency and percentage or mean and standard deviation. Comparisons of diagnosis and treatment rates by characteristics at the 1st follow-up were performed using the chi-square test or Fisher’s exact test. For time (years) until DM diagnosis and treatment, the mean, standard error, median, and 95% confidence interval (CI) were assessed using the Kaplan–Meier survival analysis. The Shapiro–Wilk test was used to confirm the normality of the data. The comparison of time from hyperglycemia to the diagnosis of DM (years) or time from DM diagnosis to treatment (years) was analyzed using nonparametric tests, including the Mann–Whitney U test and Kruskal–Wallis test, because the assumption of normality was not met. Hazard ratios (HRs) and 95% confidence intervals (CIs) were calculated using multiple Cox proportional hazards regression analysis to identify factors related to DM diagnosis and treatment. A log-minus log plot was used to confirm whether each variable met the proportional hazard assumption [24]. The prevalence of comorbidities and complications of DM at the time of diagnosis and treatment of DM was analyzed by frequency and percentage.

## 3. Results

### 3.1. Baseline Characteristics, and Diagnosis and Treatment Rates of DM

The baseline characteristics of the study participants and the diagnosis and treatment rates of DM are shown in Table 1. Of the 186 participants, 57.0% were men, 35.5% were 40–49 years old, 54.8% were current drinkers, 23.7% were current smokers, 25.8% had hypertension, 2.2% had dyslipidemia, and 7.0% had a family history of DM. Among the participants, 54.3% were obese, 21.5% had FBS levels ≥ 126 mg/dL, and 84.9% had PP2 levels ≥ 200 mg/dL. During the follow-up period, the diagnosis rate of DM was 39.8%, the treatment rate was 37.6%, and the treatment rate among participants diagnosed with DM was 94.6%.

### 3.2. Time to Diagnosis and Treatment of DM

The time period from hyperglycemia to DM diagnosis and treatment is shown in Table 2. When all participants who were not diagnosed with DM and did not receive treatment were included, the mean period to DM diagnosis was 10.87 years (median: 14.17 years), and the mean period to DM treatment was 11.34 years (median: 14.17 years) (Figure 2). The mean period from diagnosis to treatment of DM was 1.02 years (median: 0.00 years). DM was diagnosed or treated 12 years after hyperglycemia in 46.2% and 48.4% of patients, respectively. As shown in Table 3, the period to DM diagnosis showed a significant difference in the family history of DM and BMI. The time to DM was significantly shorter in patients with a family history of DM (mean: 9.42, *p* = 0.034) and BMI ≥ 25 (mean: 9.77, *p* < 0.001). The period until DM treatment showed a significant difference in the family history of DM and BMI. The duration of diabetes treatment was significantly shorter in patients with a family history of diabetes (mean: 10.75, *p* = 0.046) and BMI ≥ 25 (mean: 10.40, *p* < 0.001). The period from DM diagnosis to treatment was not significantly different according to the participant characteristics.

### 3.3. Risk Factors (Protective Factors) for DM Diagnosis and Treatment

According to the results of the multivariate analyses in Table 4, DM diagnosis was 2.41 times higher in the group with BMI ≥ 25 compared to the group with a BMI < 25 (HR = 2.41, *p* < 0.001). DM treatment was found to be affected by hypertension and BMI. Treatment of DM was 1.92 times more common in the hypertension group (HR = 1.92, *p* = 0.029) and 2.42 times more common in the group with BMI ≥ 25 than in the group with BMI < 25 (HR = 2.42, *p* = 0.001).

### 3.4. Prevalence of Comorbidities and Complications of DM at Diagnosis and Treatment

Table 5 shows the prevalence of newly occurring diabetes comorbidities and complications after hyperglycemia, observed at the time of diagnosis and treatment of diabetes. Among the participants diagnosed with DM, 36.5% (*n* = 27) were diagnosed with one or more diseases, among whom 29.7% (*n* = 22) had hypertension, 12.2% (*n* = 9) had dyslipidemia, and 2.7% (*n* = 2) had coronary artery disease. At the time of DM treatment initiation, new diseases were identified in 41.4% (*n* = 29) of patients, with hypertension being the most common at 32.9% (*n* = 23), and 2.9% (*n* = 2) had cerebrovascular disease.

## 4. Discussion

This study was conducted to determine the DM diagnosis and treatment rates, the period until diagnosis and treatment, and the prevalence at the time of diagnosis and treatment of DM in hyperglycemic adults aged ≥ 40 years using community-based cohort data followed for 18 years from 2001 to 2018. 

Based on 100 participants with confirmed hyperglycemia, approximately 40 were diagnosed with DM during the 18-year follow-up period. Additionally, among the diagnosed participants, 95% were treated with drugs and approximately 38 were treated with hyperglycemia. Approximately 60% had undiagnosed DM, and approximately 62% had not received treatment. In 2019, the global undiagnosed DM rate was 50.1%, while that in Africa and North America/Caribbean was 59.7% and 37.8%, respectively [2]. In 2013–2016, the undiagnosed DM rate in the United States was 2.6% [25]. Although Korea conducts national health checkup every 2 years, the diabetes diagnosis rate appears to be lower than that of other countries [26]. Additionally, there were differences in the design and subject selection for each study. The DM diagnosis rate in this study was lower than that in previous studies [3] using KNHANES data, which is likely to be related to differences in the study design, such as the long-term follow-up of 18 years. Moreover, participants who were not diagnosed with DM were included in the group of those that were not diagnosed with DM because hyperglycemia no longer occurred during follow-up. Accordingly, as a result of the additional analysis, 42 of the 112 undiagnosed people with diabetes did not have hyperglycemia during the follow-up period. The DM diagnosis rate increased to 51.4%, based on 144 participants who required a DM diagnosis. Of the participants diagnosed with DM, 94.6% were receiving drug treatment, and 48.6% of those with hyperglycemia were receiving drug treatment.

At the onset of hyperglycemia, all participants should be diagnosed with DM at an early stage and start pharmacological or non-pharmacological treatment according to the DM diagnostic criteria of domestic and foreign expert groups [14,15,16,17]. Therefore, it is necessary to improve diagnosis and treatment rates. DM diagnosis and treatment rates differed according to demographic and health-related characteristics. While the prevalence of obesity, hypertension, and dyslipidemia, all of which are risk factors that increase the incidence of DM complications, was high in the 60s age group [3], the diagnosis rate of DM was highest in the 50s group and lowest in the 60s group. In this regard, interventions are needed so that individuals with hyperglycemia in their 60s can be quickly diagnosed with DM and receive treatment.

The period from hyperglycemia to the diagnosis and treatment of DM requires improvement. Most participants in this study were diagnosed with DM within 10 years; however, it took more than 10 years in more than 25%, and only half were diagnosed within 6 years. According to a systematic literature review of the occurrence of diabetic retinopathy (DR), a representative diabetic small vascular complication, the annual incidence of DR differs according to the duration of DM [27]. In a study conducted in India, the average duration of DM was 5.3 years, and when followed for 4 years, the annual incidence of DR was 2.4% [28]. Additionally, in the United States, when 47% of participants who had DM for >10 years were followed up for 4 years, the annual incidence of DR was 10.4% [29]. In China, the annual incidence of DR was 2.2% when participants with an average DM duration of 5.7 years were followed for 10 years [30]. In another Chinese study, the annual incidence of DR was 12.7% when a participant with an average DM prevalence of 11 years was followed for 5 years [31], which was significantly different. Based on the results of these studies, the annual incidence of DR differed by more than fivefold between those who had had DM for <6 years and those who had had DM for >10 years. In this study, approximately 20% of the participants had been diagnosed with DM for >6 years. Additionally, new chronic diseases that were not included in the baseline were identified at the time of diagnosis in more than one-third of participants diagnosed with DM, particularly coronary artery disease, a suspected complication of DM. From these results, to prevent DM complications and improve them, it is necessary to identify issues in the process leading to DM diagnosis and treatment in people with hyperglycemia confirmed through health checkups or campaigns of health care institutions. The period from hyperglycemia to DM diagnosis in this study was shorter than that reported in previous studies [32,33]. One study reported that the period to DM diagnosis was 20 years in the population, including all adults, regardless of hyperglycemia [33], while another reported that it took 15.8 years and 11.3 years for a woman with a history of gestational DM to be diagnosed with DM after giving birth to the first child and the youngest child, respectively [32]. Thus, a direct comparison between these results should be made with caution, considering the differences in data sources and the selection of participants.

The average time to treatment initiation was 11.34 years, and most people diagnosed with DM started treatment 2 years after diagnosis. According to the study by Laiteerapong et al. [34], which analyzed data from the Kaiser Permanente Northern California DM Registry in the United States, DM control during the 1st year after exposure to hyperglycemia was strongly associated with future risk of diabetic complications and mortality, even after adjusting for glycemic control after the 1st year [34]. Compared to HbA1C < 6.5% for 1 year after diagnosis of DM, HbA1C levels of 6.5%–6.9% were associated with increased microvascular events (HR: 1.20), and HbA1C levels of 7.0%–7.9% were associated with increased mortality (HR: 1.29) [34]. In this study, hypertension was the most common comorbidity and diabetes complications when diagnosed with diabetes. In other words, subjects who did not have comorbidities or diabetes complications at baseline had already developed hypertension between hyperglycemia and diabetes diagnosis. This result supports previous studies that showed a higher incidence of diabetes in hypertensive patients than in healthy people [18], and studies that showed a 2.75-fold greater awareness of diabetes in patients with hypertension [35]. Hypertension in diabetic patients increases the risk of cardiovascular disease, and microvascular and macrovascular complications are much more prevalent in diabetic patients with hypertension than in those without hypertension [36]. Therefore, more frequent blood sugar tests are needed for patients with high blood pressure for early detection of diabetes, and education and services to increase treatment compliance are needed for blood sugar and blood pressure control. Moreover, timely treatment of DM can be beneficial in reducing the costs associated with uncontrolled DM [37]. Considering that starting DM treatment as early as possible after DM diagnosis can significantly affect the incidence of diabetic complications, mortality, and cost, it is necessary to identify the factors related to treatment after DM diagnosis and interventions to improve the treatment rate.

In this study, the protective factor for DM diagnosis in patients with hyperglycemia was BMI, and the protective factors for DM treatment were hypertension and BMI. The results of BMI as a factor influencing the diagnosis and treatment of DM differed from those of previous studies [18,38]. This can be seen as a difference in the selection criteria of the studies. Additionally, based on the study result that the factor affecting DM treatment adherence are BMI [9], and obesity is a factor influencing the prevalence of DM [38,39,40] and is known to be a major cause of chronic diseases [41], it can be interpreted that obese people are more interested in DM and are more active in DM diagnosis and treatment. Therefore, close follow-up is required in hyperglycemic patients with a BMI < 25, and early treatment of DM is required. In this study, hypertension was an influencing factor for DM treatment, which is consistent with the results of previous studies [18]. This can be considered to have a significant effect on the DM treatment rate because the prevalence of DM is high in hypertensive patients [18] and regular treatment is provided for the treatment of hypertension. Therefore, it is necessary to manage diabetes in people without high blood pressure. 

This study has several strengths. First, we used community-based cohort data to identify the period from hyperglycemia to DM diagnosis and treatment, which has rarely been studied in domestic or foreign countries. Second, factors affecting the diagnosis and treatment of diabetes in people with high blood sugar levels were identified, and the prevalence of diabetic complications at the time of diagnosis and treatment of diabetes was confirmed. Third, the validity of the data was improved by excluding data with contradictions between the DM diagnosis and drug treatment. Despite these strengths, this study also has several limitations, which necessitate caution when interpreting the results. First, the statistical power may be low owing to the small sample size because the participants were limited to those with hyperglycemia in the first follow-up. Thus, variables that are unrelated to DM diagnosis over time may change in relevance as the number of samples increases. Second, as the treatment for DM was limited to pharmacological treatment, and non-pharmacological treatments such as diet and exercise were excluded, the treatment rate may have been underestimated, and the period until the treatment of DM may have been delayed. Third, the prevalence of microvascular complications (neuropathy, nephropathy, retinopathy, etc.) at the time of diagnosis or treatment of DM could not be identified because they were not collected from baseline. Fourth, pregnancy status was not considered an inclusion criterion because the data source (KoGES) could not determine pregnancy status at the time of hyperglycemia. However, because people diagnosed with DM by a doctor at the time of hyperglycemia were excluded, patients with DM due to pregnancy may have been excluded. Finally, the generalizability of the findings to other population groups, except for the study subjects living in Ansan-Ansung, is limited.

## 5. Conclusions

This study analyzed the number of people with hyperglycemia diagnosed and treated for DM over time using community-based cohort data. The results showed that some people with hyperglycemia were not diagnosed or treated for DM or that it took a considerable period of time until they were diagnosed or treated for DM. Cerebrovascular and coronary artery diseases, suspected complications of DM, were confirmed at the time of DM treatment. Further studies are required to determine how many people with hyperglycemia discovered through health checkups or various campaigns registered and managed at public health centers or hospitals. As starting DM management as early as possible after the diagnosis of DM can greatly affect the incidence of diabetic complications, it is necessary to identify factors or problems related to treatment after the diagnosis of DM and to prepare intervention plans to improve the treatment rate.

## Figures and Tables

**Figure 1 ijerph-19-12090-f001:**
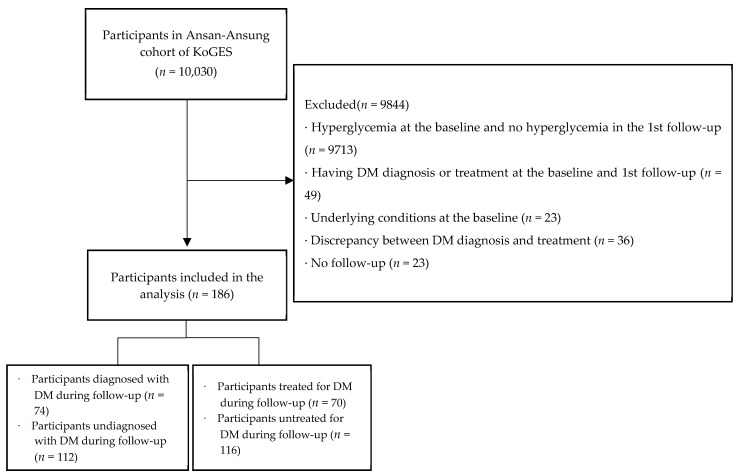
Flowchart of study participants’ selection. Note. DM—diabetes mellitus; KoGES—Korean Genome and Epidemiology Study.

**Figure 2 ijerph-19-12090-f002:**
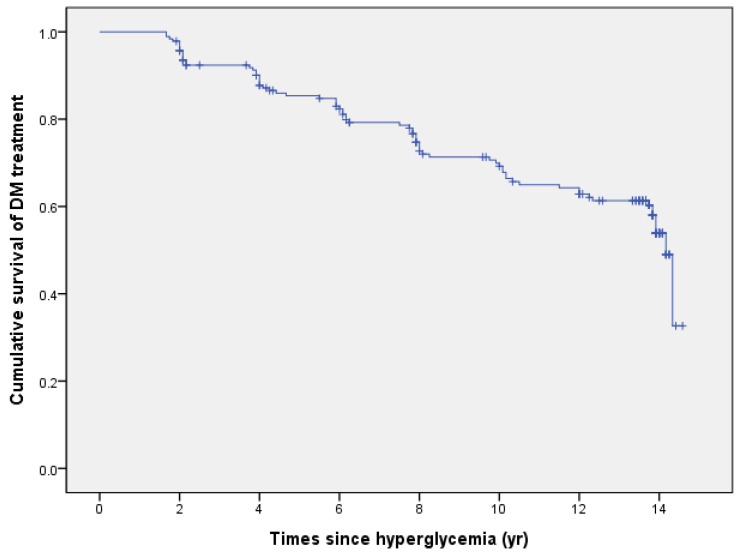
Kaplan–Meier curve for diabetes treatment over time for hyperglycemia.

**Table 1 ijerph-19-12090-t001:** Participants’ characteristics at the 1st follow-up (N = 186).

Characteristics		Total		DM Diagnosis among All		DM Treatment among All	
		*n* (%)	M ± SD	*n* (%)	*p*	*n* (%)	*p*
Sex	Male	106 (57.0)		41 (55.4)	0.723	39 (55.7)	0.785
	Female	80 (43.0)		33 (44.6)		31 (44.3)	
Age (years)	40–49	66 (35.5)	55.08 ± 8.92	25 (33.8)	0.049	24 (34.3)	0.075
	50–59	60 (32.3)		31 (41.9)		29 (41.4)	
	60–69	60 (32.3)		18 (24.3)		17 (24.3)	
Current alcohol use	No	84 (45.2)		31 (41.9)	0.466	28 (40.0)	0.272
	Yes	102 (54.8)		43 (58.1)		42 (60.0)	
Current smoking	No	142 (76.3)		55 (74.3)	0.598	52 (74.3)	0.608
	Yes	44 (23.7)		19 (25.7)		18 (25.7)	
Hypertension	No	138 (74.2)		50 (67.6)	0.093	46 (65.7)	0.040
	Yes	48 (25.8)		24 (32.4)		24 (34.3)	
Dyslipidemia	No	182 (97.8)		73 (98.6)	0.541	69 (98.6)	0.598
	Yes	4 (2.2)		1 (1.4)		1 (1.4)	
Family history of DM	No	173 (93.0)		64 (86.5)	0.005	60 (85.7)	0.002
	Yes	13 (7.0)		10 (13.5)		10 (14.3)	
Body mass index (kg/m^2^)	<25	85 (45.7)	25.08 ± 3.37	21 (28.4)	<0.001	20 (28.6)	<0.001
	≥25	101 (54.3)		53 (71.6)		50 (71.4)	
FBS (mg/dL)	<100	66 (35.5)	112.4 ± 28.42	13 (17.6)	<0.001	11 (15.8)	<0.001
	100–125	80 (43.0)		41 (55.4)		40 (57.1)	
	≥126	40 (21.5)		20 (27.0)		19 (27.1)	
PP2 ^(a)^ (mg/dL)	<140	16 (8.8)	213.22 ± 50.26	4 (5.7)	0.506	4 (6.1)	0.468
	140–199	8 (4.4)		3 (4.3)		2 (3.0)	
	≥200	158 (86.8)		63 (90.0)		60 (90.9)	
DM diagnosed	No	112 (60.2)		-		-	
	Yes	74 (39.8)		74 (100.0)		-	
DM treated	No	116 (62.4)		4 (5.4)		-	
	Yes	70 (37.6)		70 (94.6)		-	

Note. DM—diabetes mellitus; FBS—fasting blood sugar; M—mean; PP2—postprandial 2 hours glucose; SD—standard deviation. ^(a)^ Missing data were excluded.

**Table 2 ijerph-19-12090-t002:** Time to DM diagnosis and DM treatment.

Duration (Years)	Time to DM dx from Hyperglycemia	Time to DM tx from Hyperglycemia	Time to DM tx from DM dx
**For all participants**	(*n* = 186)	(*n* = 186)	(*n* = 74)
< 2.0	7 (3.8)	5 (2.7)	63 (85.1)
2.0–3.9	32 (17.2)	26 (14.0)	6 (8.1)
4.0–5.9	17 (9.1)	19 (10.2)	1 (1.4)
6.0–7.9	24 (12.9)	25 (13.4)	3 (4.1)
8.0–9.9	11 (5.9)	11 (5.9)	-
10.0–11.9	9 (4.8)	10 (5.4)	1 (1.4)
≥12.0	86 (46.2)	90 (48.4)	-
M ± SE	10.87 ± 0.36	11.34 ± 0.34	1.02 ± 0.28
Median (95% CI)	14.17 (13.92–14.42)	14.17 (13.93–14.41)	-
**For the DM diagnosed**	(*n* = 74)	(*n* = 70)	(*n* = 70)
< 2.0	6 (8.1)	4 (5.7)	62 (88.6)
2.0–3.9	21 (28.4)	14 (20.0)	5 (7.1)
4.0–5.9	10 (13.5)	12 (17.1)	1 (1.4)
6.0–7.9	14 (18.9)	13 (18.6)	2 (2.9)
8.0–9.9	7 (9.5)	7 (10.0)	-
10.0–11.9	7 (9.5)	8 (11.4)	-
≥12	9 (12.2)	12 (17.1)	-
M ± SE	6.21 ± 0.45	7.06 ± 0.47	0.63 ± 0.18
Median (95% CI)	5.92 (4.16–7.67)	6.17 (4.5–7.84)	-

Note. CI—confidence interval; DM—diabetes mellitus; dx—diagnosis; M—mean; SE—standard error; tx, treatment.

**Table 3 ijerph-19-12090-t003:** Time to DM diagnosis and DM treatment according to participants’ characteristics at the 1st follow-up.

Characteristics		Time to DM dx ^(a)^ (N = 186)			Time to DM tx ^(a)^ (N = 186)			Time from DM dx to DM tx ^(a)^ (N = 74)		
		M ± SE	MD (95% CI)	*p*	M ± SE	MD (95% CI)	*p*	M ± SE	MD (95% CI)	*p*
Sex	Male	10.71 ± 0.50	0 (0–0)	0.974	11.21 ± 0.46	0 (0–0)	0.911	1.07 ± 0.32	0 (0–0)	0.710
	Female	11.06 ± 0.53	14.17 (13.74–14.59)		11.48 ± 0.50	14.17 (13.74–14.59)		0.98 ± 0.48	0 (0–0)	
Age (years)	40–49	10.96 ± 0.60	14.33 (0–0)	0.129	11.46 ± 0.54	14.33 (0–0)	0.159	1.08 ± 0.38	0 (0–0)	0.940
	50–59	10.15 ± 0.62	12.5 (8.56–16.44)		10.75 ± 0.59	13.83 (11.87–15.80)		0.86 ± 0.38	0 (0–0)	
	60–69	11.40 ± 0.65	0 (0–0)		11.73 ± 0.61	0 (0–0)		1.00 ± 0.63	0 (0–0)	
Current alcohol use	No	10.99 ± 0.55	14.33 (0–0)	0.497	11.56 ± 0.50	14.33 (0–0)	0.305	1.27 ± 0.54	0 (0–0)	0.536
	Yes	10.71 ± 0.48	14.17 (9.19–19.14)		11.12 ± 0.45	13.92 (12.13–15.70)		0.84 ± 0.28	0 (0–0)	
Current smoking	No	11.17 ± 0.40	14.33 (13.85–14.81)	0.116	11.52 ± 0.37	14.33 (13.85–14.81)	0.127	0.81 ± 0.32	0 (0–0)	0.178
	Yes	9.62 ± 0.82	10.5 (0–0)		10.40 ± 0.75	12.00 (8.65–15.35)		1.70 ± 0.58	0 (0–0)	
Hypertension	No	11.13 ± 0.40	14.33 (13.78–14.89)	0.130	11.69 ± 0.36	14.33 (13.87–14.79)	0.062	1.27 ± 0.40	0 (0–0)	0.263
	Yes	9.96±0.76	13.92 (9.29–18.54)		10.23 ± 0.74	12.33 (9.24–15.43)		0.55 ± 0.29	0 (0–0)	
Dyslipidemia	No	10.80 ± 0.37	14.17 (13.92–14.42)	0.417	11.28 ± 0.34	14.17 (13.93–14.41)	0.459	1.04 ± 0.28	0 (0–0)	0.557
	Yes	13.44 ± 0.56	0 (0–0)		13.44 ± 0.56	0 (0–0)		0 ± 0	0 (0–0)	
Family history of DM	No	11.02 ± 0.38	14.33 (13.65–15.01)	0.034	11.42 ± 0.36	14.33 (13.65–15.01)	0.046	0.94 ± 0.31	0 (0–0)	0.362
	Yes	9.42 ± 1.19	9.92 (4.93–14.91)		10.75 ± 1.08	12.00 (9.46–14.54)		1.75 ± 0.94	0 (0–0)	
BMI (kg/m^2^)	<25	12.15 ± 0.47	14.33 (10.39–18.28)	<0.001	12.42 ± 0.44	14.33 (11.08–17.58)	<0.001	0.83 ± 0.37	0 (0–0)	0.999
	≥25	9.77 ± 0.51	12.00 (8.69–15.31)		10.40 ± 0.48	13.75 (11.58–15.92)		1.04 ± 0.34	0 (0–0)	
FBS (mg/dL)	<100	12.45 ± 0.45	-	<0.001	12.92 ± 0.37	-	<0.001	2.27 ± 1.14	0 (0–0)	0.179
	100–125	9.97 ± 0.56	12.50 (9.31–15.69)		10.38 ± 0.53	13.75 (10.64–16.86)		0.67 ± 0.22	0 (0–0)	
	≥126	9.59 ± 0.85	11.50 (5.81–17.19)		10.15 ± 0.82	13.83 (8.03–19.63)		1.04 ± 0.53	0 (0–0)	
PP2 (mg/dL)	<140	11.99 ± 1.08	-	0.478	12.44 ± 1.00	-	0.550	1.92 ± 1.92	0 (0–0)	0.498
	140–199	9.41 ± 2.25	6.00		11.04 ± 2.01	-		1.39 ± 0.58	1.83 (0–4.77)	
	≥200	10.76 ± 0.39	14.17 (13.96–14.37)		11.18 ± 0.36	14.17 (13.98–14.36)		0.96 ± 0.30	0 (0–0)	

Note. BMI—body mass index; CI—confidence interval; DM—diabetes mellitus; dx—diagnosis; FBS—fasting blood sugar; M—mean; MD—median; PP2—postprandial 2 hours glucose; SE—standard error; tx—treatment. ^(^^a)^ Censoring included.

**Table 4 ijerph-19-12090-t004:** Risk factors for DM diagnosis and treatment (N = 186).

Characteristics		DM Diagnosis				DM Treatment			
		Unadjusted HR (95% CI)	*p*	Adjusted HR (95% CI)	*p*	Unadjusted HR (95% CI)	*p*	Adjusted HR (95% CI)	*p*
Sex (ref. Male)	Female	0.99 (0.63–1.57)	0.974			0.97 (0.61–1.56)	0.912		
Age (years) (ref.40–49)	50–59	1.45 (0.85–2.45)	0.172			1.43 (0.83–2.45)	0.200		
	60–69	0.82 (0.45–1.52)	0.533			0.82 (0.44–1.53)	0.534		
Current alcohol use (ref. No)	Yes	1.17 (0.74–1.86)	0.501			1.28 (0.79–2.07)	0.308		
Current smoking (ref. No)	Yes	1.52 (0.90–2.58)	0.121			1.52 (0.88–2.61)	0.132		
Hypertension (ref. No)	Yes	1.88 (1.05–3.37)	0.035	1.66 (0.92–3.02)	0.094	2.00 (1.11–3.61)	0.021	1.92 (1.07–3.46)	0.029
Dyslipidemia (ref. No)	Yes	0.45 (0.06–3.26)	0.432			0.48 (0.07–3.48)	0.470		
Family history of DM (ref. No)	Yes	2.02 (1.04–3.95)	0.039	1.66 (0.83–3.3)	0.149	1.95 (1.00–3.81)	0.051		
BMI (kg/m^2^) (ref. <25)	≥25	2.51 (1.51–4.16)	<0.001	2.41 (1.45–4.01)	0.001	2.45 (1.46–4.12)	0.001	2.42 (1.44–4.07)	0.001
FBS (mg/dL) (ref. <126)	≥126	1.64 (0.98–2.74)	0.060			1.66 (0.98–2.82)	0.059		
PP2 (mg/dL) (ref. < 200)	≥200	1.37 (0.63–3.00)	0.431			1.55 (0.67–3.61)	0.304		

Note. BMI—body mass index; CI—confidence interval; DM—diabetes mellitus; FBS—fasting blood sugar; HR—hazard ratio; PP2—postprandial 2 hours glucose.

**Table 5 ijerph-19-12090-t005:** Prevalence of comorbidity and complications of DM at DM diagnosis and treatment.

	At the Time of DM Diagnosis (*n* = 74) *n* (%)	When to Start DM Treatment (*n* = 70) *n* (%)
Any conditions	27 (36.5)	29 (41.4)
Hypertension	22 (29.7)	23 (32.9)
Dyslipidemia	9 (12.2)	10 (14.3)
Cerebrovascular disease	0 (0)	2 (2.9)
Coronary artery disease	2 (2.7)	2 (2.9)
Myocardial infarction	0 (0)	0 (0)
Kidney disease	0 (0)	0 (0)

Note. DM—diabetes mellitus.

## Data Availability

Data in this study were from the Korean Genome and Epidemiology Study (KoGES; 4851-302), National Research Institute of Health, Korea Disease Control and Prevention Agency, Ministry for Health and Welfare, Republic of Korea.

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
