# Peer review of "Time to Diagnosis and Treatment of Diabetes Mellitus among Korean Adults with Hyperglycemia: Using a Community-Based Cohort Study"

_ijerph, 2022, doi:10.3390/ijerph191912090_

Round 1
Reviewer 1 Report
The manuscript of “Time to Diagnosis and Treatment of Diabetes Mellitus among Korean Adults with Hyperglycemia: using a Community-based Cohort Study” by Ihn Sook Jeong and Chan Mi Kang aims to determine the diagnosis and treatment rate, period until diagnosis and treatment, and prevalence at the time of diagnosis and treatment of diabetes mellitus (DM) in hyperglycemic adults aged ≥ 40 years using community-based cohort data followed up for 18 years from 2001 to 2018. The study was confirmed that some of the people with hyperglycemia were not diagnosed or treated for DM or that it took a considerable period of time until they were diagnosed or treated for DM. Cerebrovascular disease and coronary artery disease as complications of DM were confirmed at the time of DM treatment. The study has several strengths, namely community-based cohort data to identify the period from hyperglycemia to DM diagnosis, the identification of factors affecting the diagnosis and treatment of diabetes, the exclusion of data with contradictions between DM diagnosis and drug treatment. Several limitations are also indicated. The manuscript is well written and makes a significant contribution to the identification of problems related to the diagnosis of diabetes and the development of intervention plans to improve the effectiveness of treatment. The manuscript may be accepted for publication in its current form.
Minor:
Line 283 Remove the capital letter in the sentence.
Line 297 “baselin” should be replaced with “baseline”.
Author Response
Thank you for giving us the opportunity to revise our manuscript, titled “Time to Diagnosis and Treatment of Diabetes Mellitus among Korean Adults with Hyperglycemia: using a Community-based Cohort Study.” We have carefully read the editor's and reviewers’ comments and revised the overall manuscript, including the Method part. In addition, we provided a point-by-point explanation of our revisions to incorporate the comments and recommendations. We hope that these revisions improve the paper, and it is now deemed worthy of publication in International Journal of Environmental Research and Public Health by you and the reviewers.
Point 1: Line 283 Remove the capital letter in the sentence.
Response 1: Thank you for your comments. We changed the sentence as follow ;
“ Second, factors affecting the diagnosis and treatment of diabetes in people with high blood sugar levels were identified, and the prevalence of diabetic complications at the time of diagnosis and treatment of diabetes was confirmed.” (Page 9, lines 302-305)
Point 2: Line 297 “baselin” should be replaced with “baseline”.
Response 2: Thank you for your comments. We replaced the word as follow;
“Third, the prevalence of microvascular complications (neuropathy, nephropathy, retinopathy, etc.) at the time of diagnosis or treatment of DM could not be identified because they were not collected from baseline.” (Page 10, lines 314-316)

Reviewer 2 Report
- The author may like to consider to compare between DM diagnosis and treatment in the Table 1.
- N number is incorrected in the Flowchart of study. Please provide more information. The flow is not engaging to keep the reader interested to read.
- There is confusion in Table 3. N number is incorrected. it is not easy to read.
- Table 5 show shows the prevalence of diabetes comorbidities and complications, hypertension was the most common disease in diabetes comorbidities. The relationship between DM and hypertension (cardiovascular disease) needs to be discussed. These two conditions may be related. This needs to be clarified. Please discuss.
- A longer interval between diagnosis and treatment is associated with poorer prognosis among DM patient. I suggested author can stratified by various DM stages to investigate the effects of the time interval between diagnosis and treatment on overall.
Author Response
Response to Reviewer 2 Comments
Thank you for giving us the opportunity to revise our manuscript, titled “Time to Diagnosis and Treatment of Diabetes Mellitus among Korean Adults with Hyperglycemia: using a Community-based Cohort Study.” We have carefully read the editor's and reviewers’ comments and revised the overall manuscript, including the Method part. In addition, we provided a point-by-point explanation of our revisions to incorporate the comments and recommendations. We hope that these revisions improve the paper, and it is now deemed worthy of publication in International Journal of Environmental Research and Public Health by you and the reviewers.
Point 1: The author may like to consider to compare between DM diagnosis and treatment in the Table 1.
Response 1: Thank you for your comments. We added those diagnosed with diabetes and those who treated it among all subjects in Table 1 as follows, and compared the characteristics of those who diagnosed with diabetes and those who treated it;
Table 1. Participants’ characteristics at the 1st follow-up (N=186)
Characteristics |
Total |
DM diagnosis among all |
DM treatment among all |
||||
n (%) |
M±SD |
n (%) |
p |
n (%) |
P |
||
Sex |
Male |
106(57.0) |
41(55.4) |
.723 |
39(55.7) |
.785 |
|
Female |
80(43.0) |
33(44.6) |
|
31(44.3) |
|
||
Age (years) |
40-49 |
66(35.5) |
55.08±8.92 |
25(33.8) |
.049 |
24(34.3) |
.075 |
50-59 |
60(32.3) |
31(41.9) |
|
29(41.4) |
|
||
60-69 |
60(32.3) |
18(24.3) |
|
17(24.3) |
|
||
Current alcohol use |
No |
84(45.2) |
31(41.9) |
.466 |
28(40.0) |
.272 |
|
Yes |
102(54.8) |
43(58.1) |
|
42(60.0) |
|
||
Current smoking |
No |
142(76.3) |
55(74.3) |
.598 |
52(74.3) |
.608 |
|
Yes |
44(23.7) |
19(25.7) |
|
18(25.7) |
|
||
Hypertension |
No |
138(74.2) |
50(67.6) |
.093 |
46(65.7) |
.040 |
|
Yes |
48(25.8) |
24(32.4) |
|
24(34.3) |
|
||
Dyslipidemia |
No |
182(97.8) |
73(98.6) |
.541 |
69(98.6) |
.598 |
|
Yes |
4(2.2) |
1(1.4) |
|
1(1.4) |
|
||
Family history of DM |
No |
173(93.0) |
64(86.5) |
.005 |
60(85.7) |
.002 |
|
Yes |
13(7.0) |
10(13.5) |
|
10(14.3) |
|
||
Body mass index (kg/m2) |
<25 |
85(45.7) |
25.08±3.37 |
21(28.4) |
<.001 |
20(28.6) |
<.001 |
≥25 |
101(54.3) |
53(71.6) |
|
50(71.4) |
|
||
FBS (mg/dL) |
<100 |
66(35.5) |
112.4±28.42 |
13(17.6) |
<.001 |
11(15.8) |
<.001 |
100-125 |
80(43.0) |
|
41(55.4) |
|
40(57.1) |
|
|
≥126 |
40(21.5) |
20(27.0) |
|
19(27.1) |
|
||
PP2a) (mg/dL) |
<140 |
16(8.8) |
213.22±50.26 |
4(5.7) |
.506 |
4(6.1) |
.468 |
140-199 |
8(4.4) |
|
3(4.3) |
|
2(3.0) |
|
|
≥200 |
158(86.8) |
63(90.0) |
|
60(90.9) |
|
||
DM diagnosed |
No |
112(60.2) |
- |
|
- |
|
|
Yes |
74(39.8) |
74(100.0) |
|
- |
|
||
DM treated |
No |
116(62.4) |
4(5.4) |
|
- |
|
|
Yes |
70(37.6) |
70(94.6) |
|
- |
|
Point 2: N number is incorrected in the Flowchart of study. Please provide more information. The flow is not engaging to keep the reader interested to read.
Response 2: Thank you for your comments. We modified Figure 1 as follows;
Note. DM, diabetes mellitus; KoGES, Korean Genome and Epidemiology Study
Figure 1. Flowchart of study participants’ selection.
Point 3: There is confusion in Table 3. N number is incorrected. it is not easy to read.
Response 2: Thank you for your comments. We modified Table 3 as follows;
Table 3. Time to DM diagnosis and DM treatment according to participants’ characteristics at the 1st follow-up.
Characteristics |
Time to DM dx a) (N=186) |
Time to DM tx a) (N=186) |
Time from DM dx to DM tx a) (N=74) |
|||||||
M±SE |
MD (95% CI) |
p |
M±SE |
MD (95% CI) |
p |
M±SE |
MD (95% CI) |
p |
||
Sex |
Male |
10.71±0.50 |
0(0-0) |
.974 |
11.21±0.46 |
0(0-0) |
.911 |
1.07±0.32 |
0(0-0) |
.710 |
Female |
11.06±0.53 |
14.17(13.74-14.59) |
|
11.48±0.50 |
14.17(13.74-14.59) |
|
0.98±0.48 |
0(0-0) |
|
|
Age (years) |
40-49 |
10.96±0.60 |
14.33(0-0) |
.129 |
11.46±0.54 |
14.33(0-0) |
.159 |
1.08±0.38 |
0(0-0) |
.940 |
50-59 |
10.15±0.62 |
12.5(8.56-16.44) |
|
10.75±0.59 |
13.83(11.87-15.80) |
|
0.86±0.38 |
0(0-0) |
|
|
60-69 |
11.40±0.65 |
0(0-0) |
|
11.73±0.61 |
0(0-0) |
|
1.00±0.63 |
0(0-0) |
|
|
Current alcohol use |
No |
10.99±0.55 |
14.33(0-0) |
.497 |
11.56±0.50 |
14.33(0-0) |
.305 |
1.27±0.54 |
0(0-0) |
.536 |
Yes |
10.71±0.48 |
14.17(9.19-19.14) |
|
11.12±0.45 |
13.92(12.13-15.70) |
|
0.84±0.28 |
0(0-0) |
|
|
Current smoking |
No |
11.17±0.40 |
14.33(13.85-14.81) |
.116 |
11.52±0.37 |
14.33(13.85-14.81) |
.127 |
0.81±0.32 |
0(0-0) |
.178 |
Yes |
9.62±0.82 |
10.5(0-0) |
|
10.40±0.75 |
12.00(8.65-15.35) |
|
1.70±0.58 |
0(0-0) |
|
|
Hypertension |
No |
11.13±0.40 |
14.33(13.78-14.89) |
.130 |
11.69±0.36 |
14.33(13.87-14.79) |
.062 |
1.27±0.40 |
0(0-0) |
.263 |
Yes |
9.96±0.76 |
13.92(9.29-18.54) |
|
10.23±0.74 |
12.33(9.24-15.43) |
|
0.55±0.29 |
0(0-0) |
|
|
Dyslipidemia |
No |
10.80±0.37 |
14.17(13.92-14.42) |
.417 |
11.28±0.34 |
14.17(13.93-14.41) |
.459 |
1.04±0.28 |
0(0-0) |
.557 |
Yes |
13.44±0.56 |
0(0-0) |
|
13.44±0.56 |
0(0-0) |
|
0±0 |
0(0-0) |
|
|
Family history of DM |
No |
11.02±0.38 |
14.33(13.65-15.01) |
.034 |
11.42±0.36 |
14.33(13.65-15.01) |
.046 |
0.94±0.31 |
0(0-0) |
.362 |
Yes |
9.42±1.19 |
9.92(4.93-14.91) |
|
10.75±1.08 |
12.00(9.46-14.54) |
|
1.75±0.94 |
0(0-0) |
|
|
BMI (kg/m2) |
<25 |
12.15±0.47 |
14.33(10.39-18.28) |
<.001 |
12.42±0.44 |
14.33(11.08-17.58) |
<.001 |
0.83±0.37 |
0(0-0) |
.999 |
≥25 |
9.77±0.51 |
12.00(8.69-15.31) |
|
10.40±0.48 |
13.75(11.58-15.92) |
|
1.04±0.34 |
0(0-0) |
|
|
FBS (mg/dL) |
<100 |
12.45±0.45 |
- |
<.001 |
12.92±0.37 |
- |
<.001 |
2.27±1.14 |
0(0-0) |
.179 |
100-125 |
9.97±0.56 |
12.50(9.31-15.69) |
|
10.38±0.53 |
13.75(10.64-16.86) |
|
0.67±0.22 |
0(0-0) |
|
|
≥126 |
9.59±0.85 |
11.50(5.81-17.19) |
|
10.15±0.82 |
13.83(8.03-19.63) |
|
1.04±0.53 |
0(0-0) |
|
|
PP2 (mg/dL) |
<140 |
11.99±1.08 |
- |
.478 |
12.44±1.00 |
- |
.550 |
1.92±1.92 |
0(0-0) |
.498 |
140-199 |
9.41±2.25 |
6.00 |
|
11.04±2.01 |
- |
|
1.39±0.58 |
1.83(0-4.77) |
|
|
≥200 |
10.76±0.39 |
14.17(13.96-14.37) |
|
11.18±0.36 |
14.17(13.98-14.36) |
|
0.96±0.30 |
0(0-0) |
|
Note. BMI, body mass index; CI, confidence interval; DM, diabetes mellitus; dx, diagnosis; FBS, fasting blood sugar; M, mean; MD, median; PP2, postprandial 2 hours glucose; SE, standard error; tx, treatment
a)Censoring included
Point 4: Table 5 shows the prevalence of diabetes comorbidities and complications, hypertension was the most common disease in diabetes comorbidities. The relationship between DM and hypertension (cardiovascular disease) needs to be discussed. These two conditions may be related. This needs to be clarified. Please discuss.
Response 4: Thank you for your comments. We add to the discussion the relationship between diabetes and hypertension as follows;
“In this study, hypertension was the most common comorbidity and diabetes complications when diagnosed with diabetes. In other words, subjects who did not have comorbidities or diabetes complications at baseline had already developed hypertension between hyperglycemia and diabetes diagnosis. This result supports previous studies that showed a higher incidence of diabetes in hypertensive patients than in healthy people [18], and studies that showed 2.75-fold greater awareness of diabetes in patients with hypertension [35]. Hypertension in diabetic patients increases the risk of cardiovascular disease, and microvascular and macrovascular complications are much more prevalent in diabetic patients with hypertension than in those without hypertension [36]. Therefore, more frequent blood sugar tests are needed for patients with high blood pressure for early detection of diabetes, and education and services to increase treatment compliance are needed for blood sugar and blood pressure control.” (Page 9, lines 268-280)
Point 5: A longer interval between diagnosis and treatment is associated with poorer prognosis among DM patient. I suggest authors can stratified by various DM stages to investigate the effects of the time interval between diagnosis and treatment on overall.
Response 5: Thank you for your comments. We stratified and analyzed various diabetes stages to investigate the time to diagnosis and treatment of diabetes according to the DM stages (non-diabetes, pre-diabetes stage, and diabetes stage). The results are presented in Table 1 and Table 3.
Table 1. Participants’ characteristics at the 1st follow-up. (N=186)
Characteristics |
Total |
DM diagnosis among all |
DM treatment among all |
||||
n (%) |
M±SD |
n (%) |
p |
n (%) |
p |
||
Sex |
Male |
106(57.0) |
41(55.4) |
.723 |
39(55.7) |
.785 |
|
Female |
80(43.0) |
33(44.6) |
|
31(44.3) |
|
||
Age (years) |
40-49 |
66(35.5) |
55.08±8.92 |
25(33.8) |
.049 |
24(34.3) |
.075 |
50-59 |
60(32.3) |
31(41.9) |
|
29(41.4) |
|
||
60-69 |
60(32.3) |
18(24.3) |
|
17(24.3) |
|
||
Current alcohol use |
No |
84(45.2) |
31(41.9) |
.466 |
28(40.0) |
.272 |
|
Yes |
102(54.8) |
43(58.1) |
|
42(60.0) |
|
||
Current smoking |
No |
142(76.3) |
55(74.3) |
.598 |
52(74.3) |
.608 |
|
Yes |
44(23.7) |
19(25.7) |
|
18(25.7) |
|
||
Hypertension |
No |
138(74.2) |
50(67.6) |
.093 |
46(65.7) |
.040 |
|
Yes |
48(25.8) |
24(32.4) |
|
24(34.3) |
|
||
Dyslipidemia |
No |
182(97.8) |
73(98.6) |
.541 |
69(98.6) |
.598 |
|
Yes |
4(2.2) |
1(1.4) |
|
1(1.4) |
|
||
Family history of DM |
No |
173(93.0) |
64(86.5) |
.005 |
60(85.7) |
.002 |
|
Yes |
13(7.0) |
10(13.5) |
|
10(14.3) |
|
||
Body mass index (kg/m2) |
<25 |
85(45.7) |
25.08±3.37 |
21(28.4) |
<.001 |
20(28.6) |
<.001 |
≥25 |
101(54.3) |
53(71.6) |
|
50(71.4) |
|
||
FBS (mg/dL) |
<100 |
66(35.5) |
112.4±28.42 |
13(17.6) |
<.001 |
11(15.8) |
<.001 |
100-125 |
80(43.0) |
|
41(55.4) |
|
40(57.1) |
|
|
≥126 |
40(21.5) |
20(27.0) |
|
19(27.1) |
|
||
PP2a) (mg/dL) |
<140 |
16(8.8) |
213.22±50.26 |
4(5.7) |
.506 |
4(6.1) |
.468 |
140-199 |
8(4.4) |
|
3(4.3) |
|
2(3.0) |
|
|
≥200 |
158(86.8) |
63(90.0) |
|
60(90.9) |
|
||
DM diagnosed |
No |
112(60.2) |
- |
|
- |
|
|
Yes |
74(39.8) |
74(100.0) |
|
- |
|
||
DM treated |
No |
116(62.4) |
4(5.4) |
|
- |
|
|
Yes |
70(37.6) |
70(94.6) |
|
- |
|
Note. DM, diabetes mellitus; FBS, fasting blood sugar; M, mean; PP2, postprandial 2 hours glucose; SD, standard deviation. a) Missing data were excluded.
Table 3. Time to DM diagnosis and DM treatment according to participants’ characteristics at the 1st follow-up.
Characteristics |
Time to DM dx a) (N=186) |
Time to DM tx a) (N=186) |
Time from DM dx to DM tx a) (N=74) |
|||||||
M±SE |
MD (95% CI) |
p |
M±SE |
MD (95% CI) |
p |
M±SE |
MD (95% CI) |
p |
||
Sex |
Male |
10.71±0.50 |
0(0-0) |
.974 |
11.21±0.46 |
0(0-0) |
.911 |
1.07±0.32 |
0(0-0) |
.710 |
Female |
11.06±0.53 |
14.17(13.74-14.59) |
|
11.48±0.50 |
14.17(13.74-14.59) |
|
0.98±0.48 |
0(0-0) |
|
|
Age (years) |
40-49 |
10.96±0.60 |
14.33(0-0) |
.129 |
11.46±0.54 |
14.33(0-0) |
.159 |
1.08±0.38 |
0(0-0) |
.940 |
50-59 |
10.15±0.62 |
12.5(8.56-16.44) |
|
10.75±0.59 |
13.83(11.87-15.80) |
|
0.86±0.38 |
0(0-0) |
|
|
60-69 |
11.40±0.65 |
0(0-0) |
|
11.73±0.61 |
0(0-0) |
|
1.00±0.63 |
0(0-0) |
|
|
Current alcohol use |
No |
10.99±0.55 |
14.33(0-0) |
.497 |
11.56±0.50 |
14.33(0-0) |
.305 |
1.27±0.54 |
0(0-0) |
.536 |
Yes |
10.71±0.48 |
14.17(9.19-19.14) |
|
11.12±0.45 |
13.92(12.13-15.70) |
|
0.84±0.28 |
0(0-0) |
|
|
Current smoking |
No |
11.17±0.40 |
14.33(13.85-14.81) |
.116 |
11.52±0.37 |
14.33(13.85-14.81) |
.127 |
0.81±0.32 |
0(0-0) |
.178 |
Yes |
9.62±0.82 |
10.5(0-0) |
|
10.40±0.75 |
12.00(8.65-15.35) |
|
1.70±0.58 |
0(0-0) |
|
|
Hypertension |
No |
11.13±0.40 |
14.33(13.78-14.89) |
.130 |
11.69±0.36 |
14.33(13.87-14.79) |
.062 |
1.27±0.40 |
0(0-0) |
.263 |
Yes |
9.96±0.76 |
13.92(9.29-18.54) |
|
10.23±0.74 |
12.33(9.24-15.43) |
|
0.55±0.29 |
0(0-0) |
|
|
Dyslipidemia |
No |
10.80±0.37 |
14.17(13.92-14.42) |
.417 |
11.28±0.34 |
14.17(13.93-14.41) |
.459 |
1.04±0.28 |
0(0-0) |
.557 |
Yes |
13.44±0.56 |
0(0-0) |
|
13.44±0.56 |
0(0-0) |
|
0±0 |
0(0-0) |
|
|
Family history of DM |
No |
11.02±0.38 |
14.33(13.65-15.01) |
.034 |
11.42±0.36 |
14.33(13.65-15.01) |
.046 |
0.94±0.31 |
0(0-0) |
.362 |
Yes |
9.42±1.19 |
9.92(4.93-14.91) |
|
10.75±1.08 |
12.00(9.46-14.54) |
|
1.75±0.94 |
0(0-0) |
|
|
BMI (kg/m2) |
<25 |
12.15±0.47 |
14.33(10.39-18.28) |
<.001 |
12.42±0.44 |
14.33(11.08-17.58) |
<.001 |
0.83±0.37 |
0(0-0) |
.999 |
≥25 |
9.77±0.51 |
12.00(8.69-15.31) |
|
10.40±0.48 |
13.75(11.58-15.92) |
|
1.04±0.34 |
0(0-0) |
|
|
FBS (mg/dL) |
<100 |
12.45±0.45 |
- |
<.001 |
12.92±0.37 |
- |
<.001 |
2.27±1.14 |
0(0-0) |
.179 |
100-125 |
9.97±0.56 |
12.50(9.31-15.69) |
|
10.38±0.53 |
13.75(10.64-16.86) |
|
0.67±0.22 |
0(0-0) |
|
|
≥126 |
9.59±0.85 |
11.50(5.81-17.19) |
|
10.15±0.82 |
13.83(8.03-19.63) |
|
1.04±0.53 |
0(0-0) |
|
|
PP2 (mg/dL) |
<140 |
11.99±1.08 |
- |
.478 |
12.44±1.00 |
- |
.550 |
1.92±1.92 |
0(0-0) |
.498 |
140-199 |
9.41±2.25 |
6.00 |
|
11.04±2.01 |
- |
|
1.39±0.58 |
1.83(0-4.77) |
|
|
≥200 |
10.76±0.39 |
14.17(13.96-14.37) |
|
11.18±0.36 |
14.17(13.98-14.36) |
|
0.96±0.30 |
0(0-0) |
|
Note. BMI, body mass index; CI, confidence interval; DM, diabetes mellitus; dx, diagnosis; FBS, fasting blood sugar; M, mean; MD, median; PP2, postprandial 2 hours glucose; SE, standard error; tx, treatment

Reviewer 3 Report
Thank you for the opportunity to review this article. The authors tried to assess the time interval between a hyperglycemia diagnosis and the incident DM diagnosis and treatment. Although the topic is relevant, the study has considerable weaknesses.
First and foremost, the small number of individuals with hyperglycemia does not aid a credible statistical analysis.
Moreover, variables are not clearly defined, most notably, what hyperglycemia thresholds were used. Additionally, the definitions of complications and comorbidities are not sufficient.
Statistical analysis needs to explain how did you check for normality.
Strongly consider adding some graphical representation of your results.
English language should be further reviewed by a native speaker.
Author Response
Response to Reviewer 3 Comments
Manuscript Number: ijerph-1903964
Title: Time to Diagnosis and Treatment of Diabetes Mellitus among Korean Adults with Hyperglycemia: using a Community-based Cohort Study
Journal: International Journal of Environmental Research and Public Health
Thank you for giving us the opportunity to revise our manuscript, titled “Time to Diagnosis and Treatment of Diabetes Mellitus among Korean Adults with Hyperglycemia: using a Community-based Cohort Study.” We have carefully read the editor's and reviewers’ comments and revised the overall manuscript, including the Method part. In addition, we provided a point-by-point explanation of our revisions to incorporate the comments and recommendations. We hope that these revisions improve the paper, and it is now deemed worthy of publication in International Journal of Environmental Research and Public Health by you and the reviewers.
Point 1: First and foremost, the small number of individuals with hyperglycemia does not aid a credible statistical analysis.
Response 1: Thank you for your comments. Our study used data from the Korean Genome and Epidemiology Study (KoGES) [1], which can represent Korea, and this data was collected from 2001 to 2018.
It was suggested in the discussion that the analyzed statistical results should be carefully interpreted because there were not many subjects who were normal in Baseline but were found to be hyperglycemia in First follow-up. Nevertheless, the data used in this study is the only data collected in Korea over a long period of 18 years, and I think it is reliable because a number of studies[2,3] have been published using this data.
Reference
- Kim Y, Han BG, KoGES group. Cohort Profile: The Korean Genome and Epidemiology Study (KoGES) Consortium. International Journal of Epidemiology. 2017;45(2):e20. DOI: 10.1093/ije/dyv316
- Lee MY et al. Association between Serum Gamma-Glutamyltransferase and Prevalence of Metabolic Syndrome Using Data from the Korean Genome and Epidemiology Study. Endocrinology and Metabolism 2019;34(4):390-397. DOI: 10.3803/EnM.2019.34.4.390
- Wee et al. The association of asthma and its subgroups with osteoporosis: a cross-sectional study using KoGES HEXA data. Allergy, Asthma & Clinical Immunology. 2020;16(84). DOI:10.1186/s13223-020-00482-6
Point 2: Moreover, variables are not clearly defined, most notably, what hyperglycemia thresholds were used. Additionally, the definitions of complications and comorbidities are not sufficient.
Response 2: Thank you for your comments. The definition of hyperglycemic criteria was modified as follows, and the definition of diabetes comorbidities and complications was added as follows;
“ In this study, hyperglycemia was defined as abnormally high blood sugar in one or more of three glycemic indices, with glycated hemoglobin (HbA1C) ≥ 6.5%, fasting blood sugar (FBS) ≥ 126 mg/dL, or postprandial 2-hour glucose (PP2) level ≥ 200 mg/dL [14-17].” (Page 2, lines 75-78 )
“Prevalence of comorbidity was defined as an affirmative response to the survey question on the history of diagnosed hypertension or dyslipidemia and categorized into “yes” or “no”. Com-plications of DM were defined as cerebrovascular disease, coronary artery disease, myocardial in-farction, or kidney disease at the time of diagnosis or treatment of DM [6].” (Page 3, lines 116-119)
Point 3: Statistical analysis needs to explain how did you check for normality.
Response 3: Thank you for your comments. We added a sentence as follow;
“ The Shapiro–Wilk test was used to confirm the normality of the data.” (Page 2-3, line 128-129)
Point 4: Strongly consider adding some graphical representation of your results.
Response 4: Thank you for your comments. We presented the Kaplan-Meier curve for diabetes treatment over time for hyperglycemia in Figure 2 as follows;
Figure 2. Kaplan–Meier curve for diabetes treatment over time for hyperglycemia (Page 6, lines )
Point 5: English language should be further reviewed by a native speaker.
Response 5: Thank you for your comments. We had already had the abstract and manuscript reviewed once by an English proofreading institution before submission, but but we have reviewed and revised many parts according to your opinions, and received the following editorial certification.

Round 2
Reviewer 2 Report
The authors have satisfactorily addressed the concerns of this Reviewer and made the necessary changes to the manuscript.
Author Response
Point 1: The authors have satisfactorily addressed the concerns of this Reviewer and made the necessary changes to the manuscript.
Response 1: Thank you for your comments. Your comments helped our research lead to better research.
Moreover we had already had the abstract and manuscript reviewed once by an English proofreading institution before submission, but but we have reviewed and revised many parts according to your opinions, and received the following editorial certification.

Reviewer 3 Report
The authors' responses are satisfactory. However, I would like to see more detail on the definition of complications. Were they self-reported? Were official diagnostic criteria applied?
Author Response
Point 1: The authors' responses are satisfactory. However, I would like to see more detail on the definition of complications. Were they self-reported? Were official diagnostic criteria applied?
Response 1: Thank you for your comments. We revised the definition of diabetes complications as follows;
“Complications of DM were defined as an affirmative response to the survey question on the history of diagnosed cerebrovascular disease, coronary artery disease, myocardial infarction, or kidney disease at the time of diagnosis or treatment of DM [6] and categorized into “yes” or “no”.” (Page 3, lines 118-121)
